# Synthetic data in health care: A narrative review

**Aldren Gonzales**[1]*, **Guruprabha Guruswamy**[2], **Scott R. Smith**[1]

**1** Office of the Assistant Secretary Planning and Evaluation, US Department of Health and Human Services, Washington, District of Columbia, United States of America, **2** Department of Health Administration and Policy, George Mason University, Virginia, United States of America

* aldren.gonzales@hhs.gov

**Data Availability Statement:** All data are in the manuscript.

**Funding:** The authors received no specific funding for this work.

## Abstract

Data are central to research, public health, and in developing health information technology (IT) systems. Nevertheless, access to most data in health care is tightly controlled, which may limit innovation, development, and efficient implementation of new research, products, services, or systems. Using synthetic data is one of the many innovative ways that can allow organizations to share datasets with broader users. However, only a limited set of literature is available that explores its potentials and applications in health care. In this review paper, we examined existing literature to bridge the gap and highlight the utility of synthetic data in health care. We searched PubMed, Scopus, and Google Scholar to identify peer-reviewed articles, conference papers, reports, and thesis/dissertations articles related to the generation and use of synthetic datasets in health care. The review identified seven use cases of synthetic data in health care: a) simulation and prediction research, b) hypothesis, methods, and algorithm testing, c) epidemiology/public health research, d) health IT development, e) education and training, f) public release of datasets, and g) linking data. The review also identified readily and publicly accessible health care datasets, databases, and sandboxes containing synthetic data with varying degrees of utility for research, education, and software development. The review provided evidence that synthetic data are helpful in different aspects of health care and research. While the original real data remains the preferred choice, synthetic data hold possibilities in bridging data access gaps in research and evidence-based policymaking.

## Author summary

Synthetic data or data that are artificially generated is gaining more attention in the recent years because of its potential in making timely health care data more accessible for analysis and technology development. In this paper, we explored how synthetic data are being used by reviewing published literature and by looking at known synthetic datasets that are available to the public. Based on the available literature, it was identified that synthetic data address three challenges in making health care data accessible: it protects the privacy of individuals in datasets, it allows increased and faster access of researchers to health care

**Competing interests:** The authors have declared that no competing interests exist.

research data, and it addresses the lack of realistic data for software development and testing. Users should also be aware of its limitations that may include recognized risk for data leakage, dependency on imputation model, and not all synthetic data replicate precisely the content and properties of the original dataset. By explaining the utility and value of synthetic data, we hope that this review helps to improve understanding of synthetic data for different applications in research and software development.

## Introduction

Data play a significant role in advancing health care delivery, public health, research, and innovations to address barriers and improve the quality of care. When researchers and innovators have timely access to real-world data, it can inform the development of new treatment, promote evidence-based policymaking, advance program evaluation, and transform outbreak responses [1–3]. However, users continue to face different challenges to accessing original data.

Most datasets containing health information are not readily available for use because they contain confidential information about individuals. Identifiable records can't also be easily shared as organization need to comply with certain regulations, such as the Health Insurance Portability and Accountability Act of 1996 (HIPAA) in the US [4]. Researchers and analysts continue to face many barriers when accessing essential datasets. Data access requirements such as the need for data use agreements, submission and approval of full protocol, completion of data request form, ethics review approval [5,6] and cost for accessing datasets that are not in the public domain remain to be a challenge.

With more people needing access to research-identifiable records, organizations are innovating ways to make data more accessible. The generation and use of synthetic datasets can potentially address many access, privacy, and confidentiality barriers [7]. In simple terms, synthetic datasets consist entirely of, or contain a subset of, not real microdata that are artificially manufactured with or without the original data. In health care, synthetic data could be an electronic health record (EHR) dataset with patient identifiable information and other sensitive information replaced with fake data to avoid reidentification. Synthetic dataset could also contain EHR records where all the original data are synthesized to produce a completely unreal record. The formal definition and types are further discussed in the succeeding section.

While synthetic data hold great potentials to advance evidence-based policymaking, research, and innovation, challenges are still present related to its development capability and confidence in using synthetic data [8]. In addition, relatively few authors have explored the topic of synthetic data, specifically its application in the health care industry and research. While some studies have explored and used synthetic data in different ways, the discussions were very focused on the project, or the specific method used.

The goal of this review article is to serve as a guide for researchers, data entrepreneurs, and innovators to improve understanding of the utility, value, and appropriateness of synthetic data for their respective applications. The paper starts by presenting the definition and types of synthetic data. Next, synthetic data generation using various software and tools are briefly discussed. The following sections summarize use cases and description of publicly available and ready-to-download synthetic datasets. Lastly, other opportunities in using synthetic data and its limitations are highlighted in the discussion section.

## Methods

We conducted a narrative review of existing literature using PubMed and Scopus. The narrative review method was used to enable a thematic analysis of the different use cases [9]. The review was initially limited to peer-reviewed articles. These articles were identified by conducting an abstract/title search with the following terms: synthetic AND data OR dataset AND healthcare OR health care.

The articles were screened independently by two researchers. Articles presenting synthetic data development, use, and validation specific to health care delivery, public health, education, and research were included. After screening the 4,226 unique articles and reviewing 293 abstracts, 72 articles were included to identify the use cases. An additional targeted search was conducted using Google Scholar and Google Search engine to gather additional information from grey literature, including those related to the examples that were highlighted in this paper. Snowballing of references was also used to identify relevant articles from the articles that came out of the search.

## Definition and types of synthetic data

Most literature refers to the definition of synthetic data used by the US Census Bureau. It is defined as "microdata records created by statistically modeling original data and then using those models to generate new data values that reproduce the original data's statistical properties." This definition highlights the strategic use of synthetic data because it improves data utility while preserving the privacy and confidentiality of information [10]. Depending on how it is generated, synthetic datasets can come with a reverse disclosure protection mechanism for inferences about parameters in statistical models but still with adequate variables to permit appropriate multivariate analyses [11].

The term synthetic data has been widely used to characterize datasets in various synthesized forms and levels. Some argued that the term synthetic data should only be used to refer to datasets containing purely fabricated data and without any original record [12,13]. These datasets may be developed using an original dataset as a reference or modeled using statistics. However, other literature, mainly those in the census and statistics discipline, acknowledge a more diverse sub-classification of synthetic data.

In general, synthetic data can be classified into three broad categories: fully synthetic, partially synthetic, and hybrid (published originally by Aggarwal and Chu and cited by Mohan [7]). First, Rubin proposed fully synthetic data in 1993 and further developed by Raghunathan et al. [14] in 2003. It is described as a dataset that is completely synthetic in nature and doesn't contain any real data. Because there is no reality with the dataset generated, this type is considered to have a strong privacy control but low analytic value because of data loss. Second, in partially synthetic data, select variables with sensitive values and considered to be high risk for disclosure are replaced with a synthetic version of the data. Since it contains original values, the risk for reidentification is present. The idea of 'partially synthetic' data first introduced by Little in 1993 and formally named by Reiter [15] in 2003. Lastly, hybrid synthetic data is generated using both original and synthetic data. With hybrid synthetic data, "each random record of real data, a close record in the synthetic data is chosen and then both are combined to form hybrid data." It holds privacy control characteristics with high utility in comparison to the first two categories but requires more processing time and memory [7].

A more detailed spectrum of synthetic data types is described in a working paper series by the United Kingdom's Office for National Statistics (UK's ONS). The spectrum features six levels under the synthetic and synthetically-augmented dataset types [12]. According to the UK's ONS, the synthetic structural dataset (the lowest form of synthetic data and developed using

metadata only) has no analytical value and has no disclosure risk. It can only be used for very basic code testing. On the other end of the spectrum, a replica level synthetically-augmented dataset can be used in place of the real data. This dataset has high analytic value because it preserves format, structure, joint distribution, patterns, and low-level geographies. However, since it is close to the original data, it introduces more disclosure risks.

## Examples of software and tools in generating synthetic data

While published literature often refers to statistical approaches (e.g., multiple data imputation, bayesian bootstrap), the continuous development in technology has produced several tools and services to generate synthetic data programmatically. Researchers and innovators can maximize the use of software packages/libraries for R (e.g., Synthpop and Wakefield) and Python (e.g., PySynth, Scikit-learn, and Trumania) in synthesizing different types of data. Models are also available in generating synthetic images (e.g., ultrasound and computerized tomography) [16,17].

Specific to health records, there are also applications and services that users could leverage to generate synthetic data. Synthea, for example, is an open-source software package that generates high-quality, clinically realistic, synthetic patient longitudinal health records using publicly available health and census statistics, heath reports, clinical guidelines for statistical modeling [18]. MDClone's Synthetic Data Engine, on the other hand, is a commercial service that converts real EHR records into a synthetic version that is statistically comparable and maintains correlations among its variables. Health systems and universities use this synthetic data engine to accelerate data-driven medical research [19–23]. Other studies proposed the use of SynSys and Intelligent Patient Data Generator (iPDG); both are machine learning-based synthetic data generation methods for health care applications. This is not an exhaustive list of tools in generating synthetic data. As organizations explore this topic more, additional applications and services will be available.

## Uses of synthetic health data

Although the use of synthetic data can be considered as a relatively new area, few peer-reviewed and grey literature have documented its value in different areas of health care research, education, data management, and health IT development. Table 1 summarizes the different use case example where synthetic data was found to be beneficial, along with an example.

## Simulation studies and predictive analytics

Simulation and prediction research requires a large number of datasets to precisely predict behaviors and outcomes [31]. Real-world sources (e.g., from statistical agencies) have a significant advantage but are also most likely to be inaccessible to most researchers [32].

Synthetic data based on the real dataset can be used as a substitute or complement real data by allowing researchers to expand sample size or add variables that are not present in the original set. Synthetic data has been used in disease-specific hybrid simulation [33] and microsimulation for testing policy options [24,34] and health care financing strategies evaluation [35]. Studies also used synthetic data to validate simulation and prediction models [36] and to improve prediction accuracy [32].

Synthetic health records are also used in "in silico clinical trials" which refers to the development of "patient-specific models to form virtual cohorts for testing the safety and/or efficacy of new drugs and of new medical devices" [37]. The use of these datasets in silico clinical trials

Table 1. Summary of identified synthetic data use cases in health care and examples.

| Use Case | Example |
| --- | --- |
| Simulation and Prediction Research | A research project used synthetic data that is close to an external benchmark to assess the impact of policy options to visit rates, prescription, and referral rates of the 65-and-over population [24]. |
| Hypothesis, Methods, and Algorithm Testing | Experiments were conducted using publicly available and synthetic datasets to test the accuracy and robustness of the mixed-effect machine learning framework to predict longitudinal change in hemoglobin A1c among adults with type-2 diabetes [25]. |
| Epidemiological Study/Public Health Research | A simulation study used the State of California synthetic population to test the impact of isolation, home quarantine, and other interventions in reducing the number of secondary measles cases infected by the index case and the probability of uncontrolled outbreak [26]. |
| Health IT Development and Testing | The SMART Health IT sandbox contains synthetic clinical records that mimic a live EHR system environment that developers could use to test and demonstrate software applications in accessing clinical data using the SMART on FHIR platform [27]. |
| Education and Training | Oregon Health and Science University used realistic synthetic clinical cardiovascular data to teach students robust risk prediction using machine learning techniques [28]. |
| Public Release of Datasets | The Centers for Disease Control and Prevention–National Center for Health Statistics substituted select variables to prevent reidentification in the linked mortality public use files [29]. |
| Linking data | Synthetic data were used to evaluate and test the linkage quality of an algorithm to link mothers and baby records before applying it to real-world administrative data [30]. |

can also inform the clinical trial design as well as make prediction for both the population and individual level to increase the chances of success [38].

## Algorithm, hypothesis, and methods testing

Using synthetic data that reflects the content format and structure of the real data can help researchers explore variables, assess the feasibility of the dataset, and test hypotheses prior to accessing the actual dataset. Synthetic datasets can also provide another level of validation and comparison for testing methods and algorithms that would be beneficial for machine learning techniques development.

Research projects conducted experiments and have used synthetic data, public use files, and real data to verify algorithmic robustness, efficiency, and accuracy [25,39]. Another study used a publicly available synthetic claims dataset to evaluate and compare an algorithm with other models for phenotype discovery, noise analysis, scalability, and constraints analysis [40].

## Epidemiological study/public health research

Datasets for epidemiology and public health studies may be limited in size, with quality concerns, challenging to obtain because of reporting procedures and privacy concerns, and expensive due to their proprietary nature [41]. More recently, the COVID-19 pandemic underscored the importance of making data accessible for public health surveillance, clinical studies (e.g., disease prognosis, drug repurposing, and new drug and vaccine development), and policy research during a health emergency. Publication of synthesized datasets can improve the timeliness of data release, support researchers in doing real-time computational epidemiology, provide a more convenient sample for sensitivity analyses, and build a more extensive test set for improving disease detection algorithms [42].

To demonstrate its utility, a study used a synthetic model with approximately eight million virtual New York City subway riders to simulate interactions and analyze the role of subway travel in spreading an influenza epidemic [43]. During the COVID-19 pandemic, several papers have documented the potentials and actual use of synthetic data for forecasting [44], improving diagnostics using synthetic images [45], and understand risk factors [46]. Other uses include epidemiological modeling [47,48], evaluation of outbreak detection algorithms [42,49], and simulation of public health events and interventions [26].

## Health IT development and testing

Software testing is expensive, labor-extensive, and consumes between thirty to forty percent of the development lifecycle [50]. Because of the critical shortage of good test data, developers often create their own data or test with live data [51]. Using synthetic data can not only provide developers with a realistic dataset without privacy concerns, but it can also speed up the development lifecycle–saving cost, time, and labor.

Several tools are currently available such as the Michigan PatientGen tool that generates synthetic test records that are Fast Healthcare Interoperability Resources FHIR compatible. Another ready-to-download fully synthetic dataset under the SMART Sandbox mimics a live EHR production environment that developers could use for app testing and development [27].

## Education and training

Synthetic data is useful when training students in subject areas (e.g., data science, health economics) that would require students to access a large number of realistic datasets [52]. While public use and limited use files are available, important fields for analysis (e.g., county and state, birth date) are often excluded for privacy reasons.

Oregon Health and Science University documented their use of clinical cardiovascular synthetic data to teach data science students the difficulties of working with clinical and genetic covariates for prediction analytics. Aside from citing data availability issues, they used realistic synthetic data because they need a suitable dataset for novice students to use and learn on, but realistic enough to encounter difficulties in using clinical data [28].

## Public release of datasets

Releasing health datasets for public use comes with a unique challenge: preserving analytic value while ensuring the confidentiality of the records. While de-identifying microdata and data alteration can help, the probability of reidentification remains, and the alteration processes can distort the relationship of the variables [53]. Releasing partially synthesized data can mitigate disclosure risks while still allowing data users to obtain valid interferences that they could get in real data [15].

Due to high disclosure risks and to protect the confidentiality of records, the National Center for Health Statistics (NCHS) of CDC subjected linked mortality files (population survey and death certificates) to data perturbation techniques before releasing the public-use version of the dataset. Select variables that may lead to identification were replaced with synthetic values [29].

## Linking data

Linking patient records with other datasets can help answer more research questions than a single source. When combining records, data processors develop algorithms and methods to automate the process and ensure accurate linkage.

Synthetic data is widely used in testing, validating, and evaluating different data linkage methods, frameworks, and algorithms either as the primary dataset or comparison dataset [30,54–57]. A research project compared the performance of different algorithms in terms of linkage accuracy and speed using nine million synthetic records [58]. While the project also used a real dataset of more than one million records, the synthetic data provided the investigators with a larger dataset to thoroughly test the capacity and efficiency of their algorithm. The use of synthetic data is also considered one of the ways to develop 'gold standard' datasets to evaluate linkage accuracy [59].

## Examples of synthetic health datasets

Aside from project-specific datasets, there are publicly available and ready-to-download synthetic datasets that researchers, innovators, and data entrepreneurs can use for their purpose. The increasing number of these synthetic datasets is driven by the need for privacy-preserving datasets and the policies to make data available for public use. Below are examples of available datasets, databases, and sandboxes that contain synthetic data. This is not a comprehensive list but includes recently published datasets mentioned frequently in the literature review. These examples are also focused on US-specific datasets. Table 2 summarizes the data resources and their characteristics.

### CMS 2008–2010 data entrepreneurs' synthetic public use file (DE-SynPUF)

The Centers for Medicare and Medicaid (CMS) published DE-SynPUF files to make a realistic version of Medicare claims data available to the public. The 2008–2010 synthetic files contain

**Table 2. Examples of synthetic health datasets and their characteristics.**

| Synthetic Dataset | Data Owner/ Distributor | Type of Synthetic Dataset | Data Characteristics (type and quantity) | Use | Use Case Example |
|---|---|---|---|---|---|
| CMS 2008–2010 Data Entrepreneurs' Synthetic Public Use File (DE-SynPUF) | Centers for Medicare and Medicaid Services (Public domain) | Fully synthetic | 6.8 Million beneficiary records; 112 million claims records; and 111 million prescription drug events records | Data entrepreneur analysis, software and application development, research training | Used a sub-set of the DE-SynPUF dataset to test different classification algorithms to accurately predict inpatient health care expenditure [60]. |
| Synthea-Generated Datasets | MITRE Corporation | Fully Synthetic | One million longitudinal clinical synthetic patient records (SyntheticMass) | Innovation, development, education, and other nonclinical secondary uses | A pilot project used SyntheticMass data to assess whether data could be extracted from EHR through FHIR resources to support clinical trials [61]. |
| US Synthetic Household Population | RTI International | Fully synthetic | Location and descriptive sociodemographic attributes of households (116 million records) and person living in those households (300 million records) | Agent-based modeling, disease outbreak simulation, distribution of resources analysis, sociodemographic pattern recognition, and disaster planning and response. | Used the dataset to simulate the impact of different influenza epidemics and the impact of utilizing pharmacies in addition to traditional (hospitals, clinic/physician offices, and urgent care centers) locations for vaccination [62]. |
| CMS Synthetic data in Blue Button Sandbox | Centers for Medicare and Medicaid Services (inside a sandbox with access requirement) | No information | 30,000 synthetic beneficiaries with claims data (Blue Button 2.0 Sandbox) | Development and testing of applications and information systems that will need to interact with CMS data systems | Blue Button 2.0 sandbox has more than 2,000 developers using the sandbox to test data exchange [63]. |

beneficiary summary, claims, and prescription data. However, DE-SynPUF contains a smaller subset of variables of the limited use files and has undergone privacy-preserving alterations. As a result, its utility to produce reliable interference about the population has weakened and made it unsuitable for analyzing the Medicare population [64,65].

The dataset maintained the data structure, format, and metadata of the CMS limited datasets. This makes it useful for training students, for researchers at the early stage of their study in designing program codes, and for health IT innovators in testing the accuracy and safety of their systems and applications [66].

One example of how DE-SynPUF was used is a project that utilized a portion of the dataset to investigate and test various classification algorithms to help address challenges in accurately predicting which beneficiaries would increase inpatient claims [60]. Another research project used DE-SynPUF to develop a framework to detect anomalous activities in specific patient groups [67]. On the technology development and data management side, the dataset has been used to test data models [68] and methodology to query data in multiple data models [69].

### Synthea-generated datasets

One unique dataset that was generated using Synthea is the SyntheticMass. The dataset contains one million fictional but realistic residents of Massachusetts and mimics the geographic, disease rates, doctor's visits, vaccination, and social determinants of the real population [18,70].

Since the Synthea-generated datasets can be produced in FHIR formats, it is compatible with different programs and technologies for analysis and software development. Datasets from Synthea have been used in developing and testing health IT applications in FHIR environment [27,61], in teaching data science [28], and in modeling study [71].

More recently, the "Coherent Data Set" was produced using Synthea. This dataset combines multiple synthetic data forms together in a single package—familial genomes, magnetic resonance imaging (MRI) DICOM files, clinical notes, and physiological data [72].

### US synthetic household population

The US Synthetic Household Population database contains location and sociodemographic attributes representing the entire population of the US at the household and person level. The database statistically matches the real household population and accurate spatial representation, making it a viable resource for microsimulation, planning for emergency response, simulation of disease outbreaks, and assessment of public health interventions [73,74].

Although it does not contain health information, the dataset was originally developed to support a modeling study of infectious disease agents by the National Institutes of Health [75]. The dataset was used to study different models on how infectious disease spread through social contact and applied the models to analyze how seasonal illnesses spread [76,77]. Because the dataset contains geospatial variables, a study was able to simulate the impact of influenza epidemics and how administering vaccines through pharmacies in addition to the usual locations can help increase vaccination uptake [62].

### CMS synthetic data in Blue Button sandbox

CMS has sponsored different initiatives that aim to improve data access, make patient data more valuable and interoperable while minimizing the burden to health care providers [78]. Through the MyHealthEData initiative, CMS is rolling out initiatives (e.g., Blue Button 2.0) to establish standards (mostly FHIR-based) in data sharing from CMS to providers, patients, and payers [79].

CMS has published implementation guides and developed a sandbox containing 30,000 synthetic beneficiaries (in the Blue Button 2.0 sandbox) for testing purposes with over 13,000 fields from the CMS claims data warehouse mapped to FHIR [80].

## Discussion

The use cases presented in this paper highlight the utility and the value of synthetic data in health care. Considering what was covered in the review, synthetic data can address three challenges in health care data. The first is protecting the privacy of individuals and ensuring the confidentiality of records. Because synthetic data can be composed purely or mixed with "fake" data, it is harder to re-identify the records [81]. Second, it improves the access of researchers and other potential users to health data. When synthesized, datasets could be made available to a wide number of users and at a faster rate because of the minimal disclosure risk [82]. Third, synthetic data address the lack of realistic data for software development and testing. Synthetic data could be cheaper for innovators for software application testing and provide them with more realistic test data for their intended test cases [83].

Given all its advantages, leveraging synthetic data can provide great opportunities in improving data infrastructure to help address some of the emerging health challenges. The data-sharing restrictions on mental health conditions such as opioid use disorder (OUD) became barriers for researchers and public health departments [84]. Generating synthetic longitudinal records of those diagnosed with OUD and those who have died because of opioid overdose can provide researchers data that could be analyzed to study patterns, identify risks, simulate policy impacts, and evaluate the effectiveness of programs. Synthesizing datasets is also useful when studying communicable diseases and stigmatized populations where there are several barriers to data sharing, for example, people diagnosed with HIV [85].

More recently, synthetic generation gained more attention as the demand for timely and accessible data increased because of the COVID-19 pandemic. One important initiative that leveraged synthetic data is the National Institutes of Health-steered National COVID Cohort Collaborative (N3C). Aside from restricted research identifiable files, N3C also generated a synthetic version of collected EHR records to make data more available to the broader research community and citizen scientists [86]. Because of the recent development, more studies are also being conducted to validate the use of synthetic data research. Recent papers on the use of synthetic data for COVID19-related clinical research have concluded that synthetic data could be used as proxy for the real dataset and that analysis of both synthetic and real datasets would yield statistically significant results–increasing the value and utility of synthetic data.

After understanding the different uses and potential applications of synthetic data, it is also important to recognize their limitations. The promise of synthetic data is to give users a dataset with artificial variables to preserve the confidentiality of the records. However, there are still risks that some quantity of the original data could be leaked. Data leakage can happen in different ways. If data contains outliers that the model captures, characteristics are reproduced in the synthetic version of the data. For example, only three individuals are diagnosed with a rare condition in a pool of synthetic records with the usual diagnoses. Because of the unique data point, that record can be easily linked to the original data. Data leakage can also happen through adversarial machine learning techniques. When an attacker has access to the synthetic data and the generative model used to create that data, records can be identified through running inversion attacks. This could be addressed through differential privacy techniques and disclosure control evaluation [12].

Not all synthetic data replicate precisely the content and properties of the original dataset–making them less useful for generating conclusions about a population. The authors agree that

the use of synthetic data in the area of clinical research is limited at the moment [18]. However, users need to understand the source and the way the synthetic dataset was generated to evaluate its appropriateness for specific types of studies or specific stages of research. The UK's ONS synthetic data high-level spectrum provides a good illustration of the quality and usability of synthetic data based on how similar it is to the original data. ONS argues that the more the synthetic data is realistic, its analytic value increases. However, the more it is close to the original dataset, the risk for disclosure also increases [12]. Taking the CMS DE-SynPUF as an example of this limitation. The synthesizing process resulted in a significant reduction in the amount of interdependence and co-variation among the variables–making it less useful for analytics [66].

Synthetic data are also dependent on the imputation model. The quality of the resulting data is highly dependent on the model used. Models can be good with identifying statistical abnormalities in the datasets but is also vulnerable to statistical noise, such as adversarial perturbation. This may cause the model to misclassify data and produce highly inaccurate outputs. Using real-world annotated data, input into the model, and testing the output for accuracy can address this issue. Models can sometimes also focus on trends and miss on distinctive cases that the real data has. This can be an issue when using the dataset for studying and generating conclusions about a certain population contained in the synthetic dataset. An example is the limited use of the synthesized version of the CanCORS data, a large-scale health survey. Upon evaluating the synthetic data which was developed using the project's model, the researchers concluded that the dataset is only useful for preliminary data analysis purposes because of the identified issues with data relationships [87].

The limitations mentioned highlight the need for validating synthetic data and the tools/ models/algorithm used for production. Validation is needed to ensure that the synthetic dataset is comparable to the real-world data or useful for its intended purpose. The validation process can also be used to confirm the approach and evaluate if the model used worked as expected. Currently, there is no benchmark or standard for validating synthetic data [88]. Few studies have been conducted, and frameworks and approaches have been proposed to help in validating the realism of synthetic data. For example, the ATEN Framework for synthetic data generation also offers an approach to defining and describing the elements of realism and for validating synthetic data [88]. In another study, the authors compared the results derived from synthetic data generated by MDClone with those based on the real data of five studies on various topics. While the study showed that analyses conducted using synthetic data provide a close estimate of real data results in general, there are nuances observed in terms of accuracy (e.g., when a large number of patient records relative to the number of variables are used, it could yield to higher accuracy between the synthetic and real data) [23]. Another approach used to validate the realism of synthetic data is by looking at clinical quality measures. Using this approach, a study documented how synthetic data generated through Synthea could have some limitations in modeling heterogeneous outcomes and recognized the need to model additional factors that could influence deviation from guidelines and introduce variations in outcomes and quality [89]. The examples provided recognize the need for further exploration in the area of synthetic data validation, potentially towards a shared framework. Researchers and other data users should take into consideration the approach and results of validation studies in assessing if a specific synthetic dataset will be appropriate for their use.

## Conclusion

The review showed that synthetic data has the potential to bridge data access gaps. The examples cited in this review highlighted the utility of synthesized health data in different areas of health research, software development, and training. The availability of publicly available

synthetic health datasets and off-the-shelf synthetic data generators reflects the growing interest and demand for accessible data. These tools and datasets hold great potential in increasing the access of researchers, data entrepreneurs, and health IT innovators to realistic datasets while preserving the statistical relationships and protecting the confidentiality of the records. In the future, we expect more datasets with synthetic data to be released, more tools to generate synthesized data will be developed, and more users will appreciate the utility of synthetic data. Users should evaluate synthetic datasets for quality and appropriateness for their intended use. Users should be aware and take into account the limitations when using synthetic data to maximize their potentials. Future discussions and studies should examine synthetic data's validity in different use cases, establish its utility in research, validate synthetic generation tools and techniques, and promote awareness among the research and health IT community.

## Author Contributions

**Conceptualization:** Aldren Gonzales, Guruprabha Guruswamy, Scott R. Smith.

**Data curation:** Aldren Gonzales.

**Formal analysis:** Aldren Gonzales, Guruprabha Guruswamy.

**Investigation:** Aldren Gonzales, Guruprabha Guruswamy.

**Methodology:** Aldren Gonzales.

**Project administration:** Aldren Gonzales, Guruprabha Guruswamy, Scott R. Smith.

**Supervision:** Scott R. Smith.

**Writing – original draft:** Aldren Gonzales, Guruprabha Guruswamy, Scott R. Smith.

**Writing – review & editing:** Aldren Gonzales, Scott R. Smith.

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
