## [Decision Letter · Decision Letter 0]

5 Aug 2022

PDIG-D-22-00188

Synthetic data in health care: A narrative review

PLOS Digital Health

Dear Dr. Gonzales,

Thank you for submitting your manuscript to PLOS Digital Health. After careful consideration, we feel that it has merit but does not fully meet PLOS Digital Health's publication criteria as it currently stands. Therefore, we invite you to submit a revised version of the manuscript that addresses the points raised during the review process.

Both reviewers have had the chance to review the work and while they were supportive overall, the scope of the narrative review was identified as a significant limitation. Addressing this concern as well as the others identified would be necessary before further consideration.

Please submit your revised manuscript within 60 days Oct 04 2022 11:59PM. If you will need more time than this to complete your revisions, please reply to this message or contact the journal office at digitalhealth@plos.org. Please include the following items when submitting your revised manuscript:

We look forward to receiving your revised manuscript.

Kind regards,

Alistair Johnson

Section Editor

PLOS Digital Health

Journal Requirements:

1. Please amend your online Financial Disclosure statement. If you did not receive any funding for this study, please simply state: “The authors received no specific funding for this work.”

2. Please update your online Competing Interests statement. If you have no competing interests to declare, please state: “The authors have declared that no competing interests exist.”

3. Please provide a complete Data Availability Statement in the submission form, ensuring you include all necessary access information or a reason for why you are unable to make your data freely accessible. If your research concerns only data provided within your submission, please write “All data are in the manuscript and/or supporting information files.” as your Data Availability Statement.

4. We do not publish any copyright or trademark symbols that usually accompany proprietary names, eg (R), (C), or TM (e.g. next to drug or reagent names). Please remove all instances of trademark/copyright symbols throughout the text, including ® (FHIR®, Of®cial) on pages 8, 11, 14, 17, 18, 19, and 23.

Reviewers' comments:

Reviewer's Responses to Questions

**Comments to the Author**

1. Does this manuscript meet PLOS Digital Health’s publication criteria? Is the manuscript technically sound, and do the data support the conclusions? The manuscript must describe methodologically and ethically rigorous research with conclusions that are appropriately drawn based on the data presented.

Reviewer #1: Partly

Reviewer #2: Partly

2. Has the statistical analysis been performed appropriately and rigorously?

Reviewer #1: N/A

Reviewer #2: N/A

3. Have the authors made all data underlying the findings in their manuscript fully available (please refer to the Data Availability Statement at the start of the manuscript PDF file)?

Reviewer #1: No

Reviewer #2: Yes

4. Is the manuscript presented in an intelligible fashion and written in standard English?

Reviewer #1: Yes

Reviewer #2: Yes

5. Review Comments to the Author

Reviewer #1: Disclosure

I am a researcher in the synthetic health data domain, and my work is cited in this paper.

PLOS Digital Health Publication Criteria

I will note that the manuscript does not adhere to the Submission Guidelines "Manuscript Organization" for this journal. For example, there is no Materials and Methods section. A common practice for the methodology section of a review paper is to describe use of "PubMed, Scopus, etc." including search terms, how many articles the authors found, how they filtered or down-selected relevant items, counts, etc.

Writing/Editing

The phrase "While real or the original data" is awkward English. Possibly "While the original real data..."

The phrase "synthetic dataset contains all or a subset of not real (or fake) microdata" is awkward English. Suggest rephrasing, or adding commas. For example, "synthetic datasets consist entirely of, or contain a subset of, not real (or fake) microdata".

The sentence "More about formal definition and types in the section" is not complete.

General Comments

Overall, I think the article provides a good overview of the use of synthetic data in health care, including some trade-offs, and it is generally well-written. Several niche topics I thought would be missed were included, so there is some depth as well.

Consider explicitly naming or citing notable government regulations such as HIPAA or the GDPR. It is not essential, but it seems strange not to, and it could lead the interested reader to find out more.

I suggest changing "When an attacker has access to the synthetic data and the model code used to generate it" to "When an attacker has access to the synthetic data and the generative model used to create that data" as an explicit clarification, because this risk is systemic to generative adversarial models in particular and you don't need the "code" (which I interpret as "source code") you just need the resulting model.

While GANs are addressed broadly, it might be worth mentioning that these techniques are currently most useful for generating synthetic imagery (e.g., ultrasounds, MRIs). While some of the other techniques and data focus on structured longitudinal data (e.g., the CMS desynpuf data). The point is, there is a lot of variety in what synthetic health data is.

The "SyntheticUSA" project appears to be inactive for 5 years. It might be better to reference more recent work from the Synthea group. For example, the "coherent data set" (https://doi.org/10.3390/electronics11081199) or the "FDA/VHA COVID-19 challenge"

https://doi.org/10.1016/j.ibmed.2020.100007

https://precision.fda.gov/challenges/11

https://precision.fda.gov/challenges/11/results

Validation is an important topic that is covered at the end of the Discussion section. For many synthetic data sets, and some of the tools used to generate those data sets, validation is self-reported. Questions for the authors to consider discussing: How does a 3rd party validate or evaluate fit-for-purpose of synthetic data if they don't have access to the original real data nor the model used to generate that data? What level of transparency or assurance is required by different use-cases?

Final thought for author consideration: recent hot topics not covered in this review include "digital twins" and "synthetic" or "in silico" clinical trials. "Digital twin" is an umbrella term that includes many things (including modeling and simulation, big data, etc), sometimes to include synthetic health data. There is also recent work on "in silico" clinical trials using a variety of techniques, including models specifically built for that purpose and synthetic health data. Consider discussing these uses.

Reviewer #2: The topic and intentions of the paper are important. However, the narrative review is quite incomplete and the analysis is insufficiently deep, that I am unsure of the incremental value of the paper. Specific comments below:

+ The authors need to explain what is the delta compared to this paper that was published recently (or at least cite it and provide incremental information): 

https://link.springer.com/article/10.1007/s44163-021-00016-y

+ There are more vendors in this space beyond MDClone, such as Syntegra, Replica Analytics, mostly.ai, hazy, Diveplane, and more. The discussion of commercial tools is not even illustrative.

+ There are many more open source tools that are available for generating synthetic data, such as the multiple variants of GANs that have been developed for health data.

+ There are more publicly available synthetic health datasets from France, the UK, the Netherlands, and Korea. These demonstrate the diversity of synthetic data and the global adoption of the approach.

+ While it is tempting to draw general conclusions, there are a few things to note: (a) using references from the early 2000's to comment on privacy and utility risks ignores the significant work that has been in these areas over the last few years, and (b) the utility and privacy risks from synthetic data will depend on the generative models that are used and there is quite a bit of heterogeneity - a deeper analysis of these variations would give a more meaningful set of conclusions.

While I appreciate that this is only a narrative review, the gap from a good coverage of the topic is quite large that I do not think it gives a reasonable representation of the topic at all.

6. PLOS authors have the option to publish the peer review history of their article (what does this mean?). If published, this will include your full peer review and any attached files.

**Do you want your identity to be public for this peer review?** For information about this choice, including consent withdrawal, please see our Privacy Policy.

Reviewer #1: No

Reviewer #2: No

---

## [Decision Letter · Decision Letter 1]

2 Dec 2022

PDIG-D-22-00188R1

Synthetic data in health care: A narrative review

PLOS Digital Health

Dear Dr. Gonzales,

Thank you for submitting your manuscript to PLOS Digital Health. After careful consideration, we feel that it has merit but does not fully meet PLOS Digital Health's publication criteria as it currently stands. Therefore, we invite you to submit a revised version of the manuscript that addresses the points raised during the review process.

EDITOR: Please insert comments here and delete this placeholder text when finished. Be sure to:

* Indicate which changes you require for acceptance versus which changes you recommend

* Address any conflicts between the reviews so that it's clear which advice the authors should follow

* Provide specific feedback from your evaluation of the manuscript

Please submit your revised manuscript within 30 days Jan 01 2023 11:59PM. If you will need more time than this to complete your revisions, please reply to this message or contact the journal office at digitalhealth@plos.org. Please include the following items when submitting your revised manuscript:

We look forward to receiving your revised manuscript.

Kind regards,

Alistair Johnson

Section Editor

PLOS Digital Health

Journal Requirements:

Additional Editor Comments (if provided):

Thank you for revising your manuscript. We had a new reviewer who noted the lack of a validation section - which is an important point. Please add the section to your final submission. Otherwise the paper reads well and I am recommending minor revision.

Reviewers' comments:

Reviewer's Responses to Questions

**Comments to the Author**

1. If the authors have adequately addressed your comments raised in a previous round of review and you feel that this manuscript is now acceptable for publication, you may indicate that here to bypass the “Comments to the Author” section, enter your conflict of interest statement in the “Confidential to Editor” section, and submit your "Accept" recommendation.

Reviewer #1: All comments have been addressed

Reviewer #3: (No Response)

2. Does this manuscript meet PLOS Digital Health’s publication criteria? Is the manuscript technically sound, and do the data support the conclusions? The manuscript must describe methodologically and ethically rigorous research with conclusions that are appropriately drawn based on the data presented.

Reviewer #1: Yes

Reviewer #3: Partly

3. Has the statistical analysis been performed appropriately and rigorously?

Reviewer #1: N/A

Reviewer #3: N/A

4. Have the authors made all data underlying the findings in their manuscript fully available (please refer to the Data Availability Statement at the start of the manuscript PDF file)?

Reviewer #1: No

Reviewer #3: Yes

5. Is the manuscript presented in an intelligible fashion and written in standard English?

Reviewer #1: Yes

Reviewer #3: Yes

6. Review Comments to the Author

Reviewer #1: All of my comments were adequately addressed.

Reviewer #3: The paper offers a review and discussion regarding synthetic data. As the paper attempts to be comprehensive, it lacks in depth assessments of validation. To contribute to the significance of the paper, I find it particularly important to include more regard to validation, and recommend to add a section that will review validation works. 

In particular, the authors have missed an essential validation study published recently (2020), which is relevant in general for validation of synthetic data for use for medical research, and in particular concerning the MDClone system (see reference below). The paper performs a comparative validation study based on 5 research questions and datasets, and concludes important pros and cons, with no commercial bias. The paper also discusses available alternatives and approaches for generation of synthetic data. 

Please cite the paper (i) when referring MDCLone and (ii) when discussing validation. In addition, emphasize the added value of the proposed manuscript in relation to the review conducted in the introduction in that paper.

"Analyzing Medical Research Results Based on Synthetic Data and Their Relation to Real Data Results: Systematic Comparison From Five Observational Studies." Reiner Benaim et al, JMIR (2020): DOI: 10.2196/16492, PMID: 32130148, PMCID: PMC7059086.

7. PLOS authors have the option to publish the peer review history of their article (what does this mean?). If published, this will include your full peer review and any attached files.

**Do you want your identity to be public for this peer review?** For information about this choice, including consent withdrawal, please see our Privacy Policy. 

Reviewer #1: No

Reviewer #3: No

---

## [Editor Report · Decision Letter 2]

6 Dec 2022

Synthetic data in health care: A narrative review

PDIG-D-22-00188R2

Dear Mr. Gonzales,

We are pleased to inform you that your manuscript 'Synthetic data in health care: A narrative review' has been provisionally accepted for publication in PLOS Digital Health.

Best regards,

Alistair Johnson

Section Editor

PLOS Digital Health